# Length of Hospital Stay after Total Knee Arthroplasty: A Correlation Study on 1200 Patients

**DOI:** 10.3390/jcm11082114

**Published:** 2022-04-11

**Authors:** Rocco Papalia, Guglielmo Torre, Anna Maria Alifano, Erika Albo, Giuseppe Francesco Papalia, Marco Bravi, Antonio De Vincentis, Emanuele Zappalà, Biagio Zampogna, Vincenzo Denaro

**Affiliations:** 1Department of Orthopedics and Trauma Surgery, Campus Bio-Medico University of Rome, 00128 Rome, Italy; r.papalia@unicampus.it (R.P.); a.alifano@unicampus.it (A.M.A.); e.albo@unicampus.it (E.A.); emanuelezappalapio@gmail.com (E.Z.); b.zampogna@unicampus.it (B.Z.); denaro@unicampus.it (V.D.); 2Department of Physical Therapy and Rehabilitation, Campus Bio-Medico University of Rome, 00128 Rome, Italy; m.bravi@unicampus.it; 3Department of Internal Medicine and Geriatrics, Campus Bio-Medico University of Rome, 00128 Rome, Italy; a.devincentis@unicampus.it; 4Multi-Specialist Clinical Institute for Orthopaedic Trauma Care (COT), 98124 Messina, Italy

**Keywords:** total knee arthroplasty, TKA, knee, hospitalization, comorbidity, perioperative factors, elderly

## Abstract

In countries with a high average population age, total knee arthroplasty is still carried out in an inpatient setting. The roadmap to performing major surgery on an outpatient basis passes through the understanding of those perioperative features that correlate with higher lengths of hospital stay (LOS). A retrospective database of 1200 patients was reviewed for retrieving preoperative and perioperative factors including anthropometric and demographic data, comorbidities, preoperative laboratory assessment, and surgical time. Considering the LOS as a discrete series, data were analyzed by means of logistic regression with multiple univariate and multivariate models. The results showed a median length of hospital stay of 3 (IQR 3, 4) days. According to multiple univariate analysis, arterial hypertension (*p* = 0.008), diabetes mellitus (*p* = 0.028), CCI score (*p* < 0.001), ASA score (*p* = 0.006), surgical time (*p* < 0.001) and intraoperative blood loss (*p* < 0.001) were significantly associated with the duration of hospital stay in days. Moreover, preoperative hemoglobin value was inversely correlated to the LOS (*p* = 0.008). Multivariate analysis showed a significant correlation between LOS and surgical time and intraoperative blood loss. Many factors influence the permanence of the inpatient and acting on those variables, by stabilizing comorbidities and optimizing laboratory values, may reduce the overall healthcare burden.

## 1. Introduction

Total knee arthroplasty (TKA) has faced a consistent rise in volume over the last decade [1] with almost 80,000 TKA performed yearly in Italy (According to Italian National Healthcare Outcomes Program PNE-2020 Edition). An epidemiological study predicted a further increase of elective primary TKA volume of the 673% and of revision arthroplasty volume of the 601% in the United States by 2030 [2]. According to both scientific evidence and clinical experience, TKA leads to a significant improvement in quality of life and relief of pain with a satisfaction rate ranging from 80% to 100% [3]. Considering the success rate for this surgery, the massive increase in volume and the respective overload of healthcare facilities for the assistance burden that arthroplasty surgery brings, a great effort is being paid worldwide to reduce the length of hospital stay after surgery and to achieve the objective of performing total or partial joint arthroplasty on an outpatient basis [4,5]. Evidence has been developed on the increase of complications after knee arthroplasty in outpatient setting [6]. Thus, the transition to the outpatient basis is still debated in many European countries, while in United States is a concrete reality in most arthroplasty centers [7]. Several perioperative factors and possible early complications [8] influence the immediate postoperative outcome, playing a relevant role in the patient’s needs after the surgery [9]. A great debate is present in literature, concerning the safety and efficacy of early discharge and of outpatient setting for total joint arthroplasty, though the evidence is still not definite (grade B) in supporting these scenarios in routine orthopedics [4]. Furthermore, several studies are investigating the role of elderly frailty in influencing the postoperative course after knee arthroplasty [10] and the underlying role of a multidisciplinary approach to the perioperative management of these patients. As for all major surgical procedures, the presence of several comorbidities makes mandatory the stabilization of the general clinical conditions of the patient before he or she can undergo TKA. On the other hand, the impact of surgery on stable comorbidities may be detrimental, leading to an impaired general status, that must be optimized before discharge either at home or at the rehabilitation department. Given the complexity of such a relevant balance between success, increased surgical volumes and individualized patient care, the endpoint of outpatient surgery needs heavy work to be pursued in several European countries, especially in those with higher mean population age. At our institution, outpatient TKA is not an option, and all cases are managed with overnight stay for a few days after the surgery. A first step ahead could be represented by individuating those factors that correlate with an increased length of hospitalization after TKA, for a better understanding of crucial patient characteristics and surgical features to be optimized. The purpose of the present investigation was to identify potentially predictive perioperative factors correlated with the length of hospital stay in the orthopedic ward in a large cohort of patients who underwent elective primary TKA with overnight admission. Specifically, our goal was to evaluate how comorbidities, length of surgical procedure, and preoperative laboratory values could correlate with the hospitalization length. Although several papers have already been published on the topic, this represents a first attempt to collect in a single multivariable model all the factors that are known to be relevant to predict the LOS after TKA.

## 2. Materials and Methods

### 2.1. Study Design

Electronic medical records at Campus Bio-Medico University Hospital, in Rome, Italy, were retrospectively evaluated. Medical records of patients that underwent TKA since the 1st of January 2016 to the 31st of December 2018 were screened. The study was approved by the Institution’s Ethical Committee (Protocol number: 32/19 OSS ComEt CBM). Informed consent to the use of personal health-related data for research purposes was signed by all patients. The study was conducted in respect of the principles of the declaration of Helsinki for research on human subjects. The STROBE statement was adopted for a correct reporting check.

### 2.2. Population

Inclusion criteria were adult subjects (no age range limiting the selection) that underwent elective primary TKA surgery; diagnosis of primary or secondary osteoarthritis of the knee; medical record available within electronic database of the institution; complete medical records. Exclusion criteria were the following: revision surgery, oncologic patients, one-stage bilateral procedures, no informed consent given for data collection. Surgical procedures were carried out at our institution by two experienced surgeons, with a minimum 15 years of experience in knee joint replacement. The sselection process is summarized in Figure 1. No comorbidity prevented the patient to be included for review, except for oncologic comorbidities (according to exclusion criteria). All the patients underwent the same perioperative thromboembolic prophylaxis with bilateral elastic stockings, low molecular weight heparin, at a dose calculated according to weight. Furthermore, the standard antibiotic prophylaxis with II generation cephalosporin was administered (2 g preoperative, 1 g every 5 h for three times, postoperatively). For those with known hypersensitivity reaction to penicillin and cephalosporins, clindamycin (500 mg before the surgery and 500 mg 12 h after surgery). In the first postoperative day, patients were instructed and assisted by physiotherapists for performing active and passive range of motion exercises (with and without Continue Passive Motion machine) and were able to sit on the bed. In the second day, at morning they were instructed on walking with crutches with tolerance weight bearing. Discharge to a rehabilitation facility was provided as soon as patients were able to walk with two crutches for at least 5–7 min, and pain was controlled. Hemoglobin (Hb) level was checked daily to ensure the patient was discharged with an acceptable Hb level (stable or with ascending trend).

### 2.3. Data Extraction

Of the data available in the medical records of the included patients, after accurate review of the literature, consensus was reached within a multidisciplinary team of investigators (G.T., R.P., B.Z., A.M.A., and M.B.) about data to be included in the analysis. Demographic data included sex and age. Among comorbidities, patient’s records were screened for presence of diabetes mellitus, arterial hypertension, dyslipidemia, heart failure, coronary disease, chronic obstructive pulmonary disease (COPD), and chronic kidney disease (CKD). The American Society of Anesthesiologists (ASA) score was also extracted. Furthermore, information about smoking habits were retrieved (smokers or smoke in the past). Preoperative whole blood count, glycemia, and serum creatinine data were also collected. Surgical details included: operative time, patellar resurfacing, and intraoperative blood loss. Data on the length of postoperative orthopedic ward stay were retrieved.

### 2.4. Data Processing

As in a previous published paper [11] the LOS was processed as a discrete variable (days of stay after surgery), obtained by the difference in nights between the date of discharge and the day of surgery. LOS was also transformed into a dichotomous variable, where the median value of the discrete series was used as a cut-off to identify a short (minor or equal to the median) or long (major than the median) stay. The Charlson Comorbidity Index (CCI) [12] was computed to represent each patient’s global burden of comorbidities. This index was processed as a discrete 5-levels variable and as a dichotomous variable with a cutoff value of 1, consistently with previous literature [13].

### 2.5. Statistical Analysis

Database consistency and completeness was assessed before proceeding with the planned analysis. Data series had a maximum incompleteness of 11%. Missing data in continuous series were addressed through the imputation of the mean, in discrete or binomial distribution, the median was used for imputation. Sensitivity analysis showed that no difference in core distribution of the series changed significantly after imputation. Continuous variables were shown as mean with standard deviation (SD), while discrete variables through the median and interquartile range (IQR). Categorical Binomial variables were presented as absolute numbers and percentages. Length of hospital stay after surgery was defined as dependent variable, distribution and variance were checked, to choose the best regression model. The association between clinical, laboratory, and surgical factors and the length of hospital stay was explored by means of multiple univariate negative binomial regression models and expressed as incident risk ratio (IRR) with 95% confidence intervals (95%CI). A multivariable negative binomial model was also fitted including as independent variables all factors found associated with *p* < 0.10 at univariate analysis. Stepwise selection with backward deletion was carried out, to define a model with all-significant independent predictors. The best-fit negative binomial multivariate model was subsequently fitted in logistic regression, using the dichotomic transformation of LOS as dependent variable. The output was reported as odds ratio with 95%CI. The significance threshold was conventionally set with *p* < 0.05. All analyses were conducted with STATA software for statistical computing for Mac (version 12, Stata Corp, College Station, TX, USA).

## 3. Results

A cohort of 1256 patients satisfied inclusion criteria and medical records were obtained for data collection and analysis. There were 414 (32.9%) males, and the mean age was 69.8 ± 8.4 years. Mean BMI was 29.4 Kg/m^2^, and 127 (10.1%) patients were smokers at the moment of surgery. The mean surgical duration was 54.4 ± 14.4 min. The median length of hospital stay was 3 (IQR 3, 4) days. The median CCI was 1 (IQR 0, 1), the incidence of specific comorbidities is summarized in Table 1. Preoperative lab values are summarized in Table 1. Preoperative Charlson’s Comorbidity Index and ASA score are shown in Table 2.

Regression models are reported in Table 3. According to multiple univariate analysis (negative binomial model), the presence of arterial hypertension (IRR = 1.08, *p* = 0.008) or of diabetes mellitus (IRR = 1.12, *p* = 0.028), the CCI score (IRR = 1.06, *p* < 0.001), a CCI higher than 1 (IRR = 1.07, *p* = 0.03), the ASA score (IRR = 1.18, *p* = 0.006), the surgical time (IRR = 1.002, *p* < 0.001), and intraoperative blood loss (IRR = 1.001, *p* < 0.001) were directly correlated to LOS in days. Conversely, the preoperative hemoglobin value (IRR = 0.97, *p* = 0.008) was inversely correlated to the LOS.

Multivariate models with best fit showed that either using the negative binomial (NB) regression or the logistic (Log) regression, the only factors significantly associated with LOS were surgical time (NB: IRR = 1.001, *p* = 0.011, Log: IRR = 1.0005, *p* = 0.012) and intraoperative blood loss (NB: IRR = 1.006, *p* = 0.001, Log: IRR = 1.001, *p* = 0.04).

## 4. Discussion

The present study retrospectively investigated factors that influenced the length of hospital stay in the orthopedic ward after elective primary TKA to optimize known factors yielding a longer hospitalization. Regarding comorbidities, our findings suggest that patients with arterial hypertension, diabetes mellitus, or a CCI higher than 1 have a statistically increased risk of longer LOS. Furthermore, surgical time and intraoperative blood loss showed a significant association to LOS both in the univariate and multivariate analysis. Moreover, preoperative hemoglobin values presented inverse correlations to the LOS. In the current literature, several studies highlighted the need to reduce hospitalization, adverse events, and hospital readmission rates, focusing on the complexity of the patient. Concerning comorbidities, it has been reported that type 2 diabetes can be considered a risk factor for in-hospital postoperative complications, potentially increasing the LOS [14]. In the data available for our study, there was no information about the type of diabetes, and this prevented us from specifically analyzing the role of type 1 and type 2 diabetes. This could be a relevant focus of future studies. However, a paper by Marchant et al. reported that independently from diabetes type, the uncontrolled hyperglycemia increases the odds for postoperative complications and total LOS [15]. Similarly, a recent paper by Na et al. showed that independently from the type of diabetes, the 90 days readmission was increased after elective knee arthroplasty [16]. Hypertension represents another independent risk factor, well described in literature. Revision TKA in octogenarians has been reported to have an increased LOS if the patient has hypertension, along with an increased risk of developing complications [17]. Furthermore, optimization of blood pressure values in the immediate postoperative period, showed a significant decrease of mobilization-induced adverse events [18]. However, another recent paper showed a protective role of hypertension (1 day shorter LOS, *p* < 0.001) after elective TKA [19], though using a different statistical methodology.

Generally, males had a significantly shorter stay, while the female gender has historically been associated with longer LOS [20]. In consideration of the multidisciplinary nature of the topic, a team of anesthesiologists analyzed a cohort of patients with suspected obstructive sleep apnea undergoing total knee arthroplasty. They underlined the importance of the preoperative value of serum CO_2_ as a factor that could determine prolonged episodes of desaturation postoperatively and consequent relationship with longer length of stay. In fact, patients with a preoperative serum CO_2_ of 30 mmol/L or greater had longer desaturation episodes significantly on postoperative day 0 compared with CO_2_ of less than 30 mmol/L [21]. Furthermore, albuminemia has been evaluated as a factor associated to an increased cost of hospitalization after knee and hip arthroplasty, showing that this metric was increased by 16.2% in patients with preoperative hypoalbuminemia [22]. Similarly, Bohl et al. showed an increased risk for postoperative complications in patients with hypoalbuminemia, along with increased LOS [23]. A study on the predictive factors of hospitalization length focused on purely anesthesiologic features (ASA score) and elements of orthopedic competence (wound exudate, and ROM at day 0) to determine the treatment success. While it is predictable that ASA is related to a lower risk of complications, great importance is given to fast track and surgical wound management to reduce the LOS [24]. In accordance with our results, a study by Monsef et al. on 516 primary TKA found that preoperative optimization of Hb level provided a significantly reduced transfusion rate, length, and cost of hospital stay [25]. In the same way, another study analyzed the impact of comorbidities relatively on the length of stay for total knee arthroplasty in a cohort of 2009 patients after TKA. In this case, the only variable that significantly predicted LOS was age (*p* < 0.001), while other factors such as gender, smoking status, venous thromboembolism history, body mass index, and diabetes status were not significant predictors [26]. The perfect definition of multidisciplinary approach is represented by the ERAS (enhanced recovery after orthopedic surgery) protocol that investigated the effect pathway on patient hospitalization. A substantial improvement in post-surgical recovery and a decreased readmission rate was reported after the application of the ERAS pathways [27]. However, the ERAS protocol is challenging to implement in clinical practice and the patients of the present study were not managed with ERAS protocol. Among the element investigated in the analyzed literature, the BMI (body mass index) was partially considered, if not in correlation with the ASA score [24]. Conversely, a retrospective study focused on obesity detected the association between different classes of BMI and postoperative adverse outcomes. In this study, 1665 TKA were performed in patients classified according to their body weight in different categories. The risk, layered by class, resulted in lower postoperative complications in obese patients than the average weight category. These controversial results were interpreted cautiously because using only BMI as a covariate prevents the analysis of confounding factors, yielding a frail study conclusion [28]. Jansen et al., in their prospective study, highlighted as the negative effect of preoperative hypovitaminosis D on postoperative scores after TKA and a longer stay in hospital. In fact, 24% of the study population (33 patients of 138) that had vitamin D deficiency showed both significant worse WOMAC score at long term follow-up (*p* = 0.04; +5.0) and significant longer stay in hospital (*p* = 0.03; +1.0 day). Especially higher age and female sex were associated with significantly lower functional outcome scores [29]. On the same topic a study of the HCAHPS survey in American patients, published in 2019, found a correlation between age (a 0.3% decrease in top-box response rate), race (African American correlated with a 5.6% increased rate for top-box response), marital status (divorced/separated status resulted in a significant 5.4% decrease) and postoperative outcomes after TKA. Conversely, gender, BMI, smoking status, insurance type, and discharge disposition were not significantly correlated with HCHAPS top-box response rate [30]. It is worth to consider that in last years, the North American healthcare system faced a definite and significant shift towards the outpatient management for major surgery, including TKA, in order to lower the financial pressure on the patients [31]. Although this risk–benefit balance may seem to be valid only in economic aspects, European countries are trying to keep up with this trend. In Italy and in other countries, where public healthcare still provides full coverage for inpatient surgery, the priority for developing outpatient management strategies is low. However, in a perspective of savings for the individual healthcare facility, the reduction of the LOS is still a challenge.

The strength of the present study is the large cohort, the evaluation of only primary elective TKA, and electronic database records retrieved by an independent investigator not involved in the study. Furthermore, we investigated all the preoperative factors potentially relevant to hospital stay length, with a multivariable model comprehensive of all these factors, for the first time in literature, according to the authors knowledge. However, the study is not free from limitations, including the retrospective design with the imputation of missing data, the unbalanced distribution of most of the data series (positive skewness) and the absence of relevant data such as blood transfusion rate and details on severity and type of diabetes.

## 5. Conclusions

In conclusion, among the several analyzed variables, our study showed a significant association of LOS with the presence of arterial hypertension or of diabetes mellitus, the CCI score, the ASA score, the surgical time and intraoperative blood loss, at univariate regression. Moreover, surgical time and intraoperative blood loss were the only factors significantly associated with LOS also at multivariate regression. There is a need to optimize patient’s preoperative hemoglobin value to improve the general conditions of the patient and yield a better surgical response. However, the road to outpatient joint arthroplasty for aged patient population is still long.

## Figures and Tables

**Figure 1 jcm-11-02114-f001:**
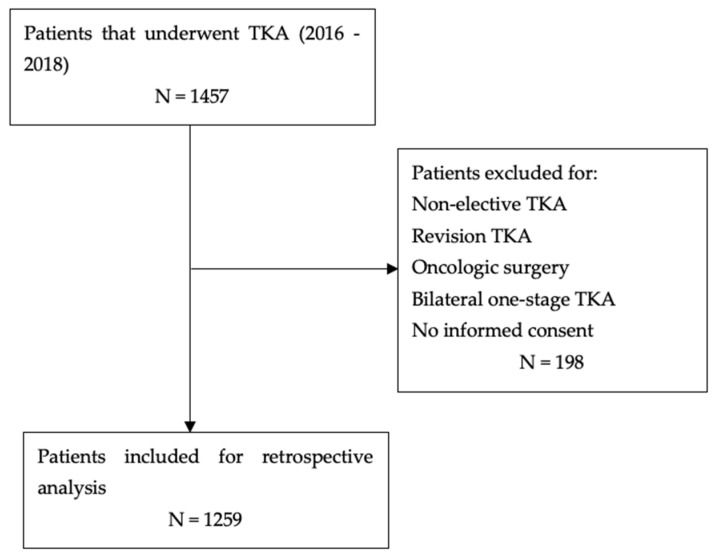
Medical records selection flow-diagram.

**Table 1 jcm-11-02114-t001:** Descriptive statistics of study variables.

Variable, Measure	Unit	Summary
** *Demographic, anthropometric:* **		
Age, mean (SD)	Years	69.8 (8.4)
Male sex, *n* (%)	*n*	414 (32.9)
BMI, mean (SD)	Kg/m^2^	29.4 (4.5)
** *Comorbidities:* **		
Diabetes Mellitus, *n* (%)	*n*	132 (10.4)
Arterial hypertension, *n* (%)	*n*	685 (54.48)
Dyslipidemia, *n* (%)	*n*	228 (18.1)
Heart failure, *n* (%)	*n*	7 (0.6)
Coronary disease, *n* (%)	*n*	12 (0.95)
COPD, *n* (%)	*n*	15 (1.2)
CKD, *n* (%)	*n*	1 (0.1)
Smokers, *n* (%)	*n*	127 (10.1)
Smoke in the past, *n* (%)	*n*	193 (15.3)
** *Preoperative laboratory values:* **		
BFG, mean (SD)	mg/dL	105.4 (19.1)
Hb, mean (SD)	g/dL	13.9 (1.41)
WBC, mean (SD)	U/microl	7051.1 (1004.4)
Creatinine, mean (SD)	mg/dL	0.83 (0.25)
** *Perioperative factors:* **		
Operation time, mean (SD)	Minutes	54.4 (14.4)
Patella resurfacing, *n* (%)	*n*	351 (27.9)
Intraoperative blood loss, mean (SD)	ml	183.1 (102.1)
** *Hospitalization length (LOS):* **		
Hospitalization length, median (IQR)	days	3 (3,4)
Patients with LOS longer than the median, *n* (%)	*n*	521 (41.4)

**Table 2 jcm-11-02114-t002:** Charlson’s Comorbidity Index and ASA score.

Charlson’s Index	*n* (%)
0	507 (40.3)
1	477 (37.9)
2	219 (17.4)
3	52 (4.1)
4	4 (0.3)
**ASA Score**	***n* (%)**
1	0 (0)
2	1203 (95.6)
3	56 (4.5)
4	0 (0)

**Table 3 jcm-11-02114-t003:** Regression Models.

Variable	Univariate Negative Binomial		Multivariate Negative Binomial (Best Fit) Chi^2^ = 35.2			Multivariate Logistic Chi^2^ = 1.67
	IRR (95% CI)	*p*-Value	IRR (95% CI)	*p*-Value	OR (95% CI)	*p*-Value
** *Demographic, anthropometric:* **						
Age	1.00 (0.99, 1.0)	0.229				
Male sex	1.04 (0.97, 1.1)	0.26				
BMI	1.00 (0.99, 1.01)	0.492				
** *Comorbidities:* **						
Diabetes Mellitus	1.12 (1.01, 1.24)	0.028 *	1.10 (0.99, 1.23)	0.07	1.35 (0.81, 2.22)	0.239
Arterial hypertension	1.08 (1.01, 1.14)	0.009 *	1.03 (0.96, 1.22)	0.4	1.34 (0.99, 1.8)	0.052
Dyslipidemia	1.08 (1.01, 1.15)	0.015 *				
Heart failure	1.26 (0.99, 1.61)	0.058				
Coronary disease	1.11 (0.87, 1.41)	0.400				
COPD	1.13 (0.92, 1.39)	0.257				
Smokers	0.96 (0.89, 1.04)	0.415				
Smoke in the past	0.97 (0.89, 1.07)	0.630				
CCI	1.06 (1.02, 1.09)	<0.001 **				
CCI > 1	1.07 (1.01, 1.13)	0.03 *	0.96 (0.87, 1.07)	0.456	0.85 (0.57, 1.3)	0.448
ASA	1.18 (1.05, 1.32)	0.006 *				
** *Preoperative laboratory values:* **						
BFG	1.00 (0.99, 0.011)	0.289				
Hb	0.97 (0.95, 0.99)	0.008 *	0.98 (0.96, 1.01)	0.143	0.91 (0.83, 1.01)	0.065
WBC	0.99 (0.99, 1)	0.791				
Creatinine	1.09 (0.94, 1.27)	0.230				
** *Perioperative factors:* **						
Operation time	1.002 (1, 1.003)	<0.001 *	1.001 (1, 1.002)	0.011 *	1.006 (1.002, 1.01)	0.001 *
Patellar resurfacing	1.02 (0.96, 1.08)	0.365				
Intraoperative blood loss	1.001 (1, 1.001)	<0.001 *	1.0005 (1, 1.001)	0.012 *	1.001 (1, 1.003)	0.04 *

IRR: incident risk ratio; OR: odds ratio. * Significant; ** highly significant.

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
