# Peer review of "Length of Hospital Stay after Total Knee Arthroplasty: A Correlation Study on 1200 Patients"

_jcm, 2022, doi:10.3390/jcm11082114_

Round 1
Reviewer 1 Report
The manuscript “Length of hospital stay after total knee arthroplasty: a correlation study on 1,200 patients” by Rocco Papalia, Guglielmo Torre, Anna Maria Alifano, Erika Albo, Giuseppe Francesco Papalia, Marco Bravi, Antonio De Vincentis, Emanuele Zappalà, Biagio Zampogna, Vincenzo Denaro aimed to evaluate how comorbidities, length of surgical procedure, and preoperative laboratory values could correlate with the hospitalization length.
Below are my comments and remarks regarding the article:
1. Primary and secondary degenerative changes were not addressed as to whether there were differences in the results
2. Diabetes and hypertension should be taken into account whether it is under control or not, and the duration of these diseases. it was not described whether diabetes was insulin-dependent or not.
3. The acronym CCI should be expanded in Table 3
4. Were there patients with ASA 1 or 4 in such a large population?
5. LOS is multifactorial - and the study analyzed only its length without eliminating factors, e.g. non-medical influencing it
6. The discussion referred to ERAS - the authors of the study did not report whether the THA procedures were compliant or not with ERAS
7. The discussion is not very clear - without a more detailed discussion of the test results, ie diabetes and hypertension
Author Response
Dear Reviewer,
thank You for your suggestion.

Reviewer 2 Report
Hello, thank you for your submission to the Journal. The substance of the work is important, as the rest of the world has much catching up to do with the United States with regards to long inpatients stays after a benign procedure such as the primary total knee arthroplasty. I find it amusing that the average stay for TKAs in this facility is three days. The goal of showing optimization for certain variables to bring down this number is to be applauded. However, there are a few major issues that require addressing, one of which is the OARA score which should have been a part of the analysis. If not the OARA score, some mention of a system that allows for stratification of outpatient/inpatient patient selection would be ideal here. In other words, what is the solution for figuring out who should be inpatient versus outpatient? If this is not the aim of this study, bur rather to just demonstrate how to reduce the number of days in the average stay, this omission is understandable. However, outpatient status for TKA is the norm, and should have more about this in the discussion at the very least.
Some individual points:
1) Throughout the manuscript, there appears to be capitalization of words that should not be capitalized. Please have an editor go through and fix this. For example, Line 18 "Stay"; Line 92 "Clindamycin". Abbreviated words do not get capitalized, i.e. the first mention of "Total Knee Arthroplasty" should be "total knee arthroplasty".
2) Is CKF the acceptable disease name? Or is it CKD "chronic kidney disease". Please double check.
3) There are only 20 references here, there is much more in the literature regarding the optimization of patient variables peri-TKA.
4) Was the CCI age adjusted? If not, why?
5) Diabetes mellitus must be described as type I or type II, please indicate which was analyzed. I hope these were not grouped in analysis.
6) INR and albumin levels have been analyzed as another possible metric for optimization, please reference and be a little more thorough in the references.
7) Sub scales for ASA? Not just ASA, but how about ASA-PS? etc.
Author Response

(The authors gave the same response as above.)

Round 2
Reviewer 1 Report
Only some corrections were made to the suggestion in the discussion. The remaining sections have not been corrected.
Author Response
Reviewer 1
Only some corrections were made to the suggestion in the discussion. The remaining sections have not been corrected.
Answer:
Dear Reviewer,
We are sorry that you did not evaluate the corrections that we have made in the discussion. We expanded the discussion accordingly to your suggestion with the results regarding diabetes and hypertension, from lines 183 to 198. Moreover, we inserted a paragraph about the role of hypoalbuminemia as a risk factor, from lines 207 to 211. However, all the additions in the discussion are reported in bold. For any doubt, please feel free to contact us again.